# Epidemiology and Clinical Aspects of Malignant Pleural Mesothelioma

**DOI:** 10.3390/cancers13164194

**Published:** 2021-08-20

**Authors:** Fraser Brims

**Affiliations:** 1Curtin Medical School, Curtin University, Perth, WA 6845, Australia; fraser.brims@curtin.edu.au; 2Department of Respiratory Medicine, Sir Charles Gairdner Hospital, Perth, WA 6009, Australia; 3National Centre for Asbestos Related Diseases, Institute for Respiratory Health, Perth, WA 6009, Australia

**Keywords:** mesothelioma, asbestos, epidemiology, pleural disease

## Abstract

**Simple Summary:**

Mesothelioma is a cancer of the lining of the lungs caused by breathing in asbestos fibres. Asbestos was widely used in industry in the last century in most developed countries and is still present in many older buildings to this day. There is no known safe level of asbestos exposure. Symptoms of mesothelioma can include worsening breathlessness, chest pain and loss of weight. There is no cure, and the treatment of mesothelioma is limited, although there have been some recent improvements in therapy. Survival is very variable although most people live for around one year after diagnosis. Efforts to improve and maintain the quality of life for patients with mesothelioma remain a priority.

**Abstract:**

Mesothelioma is a cancer predominantly of the pleural cavity. There is a clear association of exposure to asbestos with a dose dependent risk of mesothelioma. The incidence of mesothelioma in different countries reflect the historical patterns of commercial asbestos utilisation in the last century and predominant occupational exposures mean that mesothelioma is mostly seen in males. Modern imaging techniques and advances in immunohistochemical staining have contributed to an improved diagnosis of mesothelioma. There have also been recent advances in immune checkpoint inhibition, however, mesothelioma remains very challenging to manage, especially considering its limited response to conventional systemic anticancer therapy and that no cure exists. Palliative interventions and support remain paramount with a median survival of 9–12 months after diagnosis. The epidemiology and diagnosis of mesothelioma has been debated over previous decades, due to a number of factors, such as the long latent period following asbestos exposure and disease occurrence, the different potencies of the various forms of asbestos used commercially, the occurrence of mesothelioma in the peritoneal cavity and its heterogeneous pathological and cytological appearances. This review will describe the contemporary knowledge on the epidemiology of mesothelioma and provide an overview of the best clinical practice including diagnostic approaches and management.

## 1. Introduction

Malignant pleural mesothelioma (MPM) is a cancer caused by exposure to asbestos that only became widely recognised in the second half of the last century. The widespread utilisation of asbestos in developed countries in the last century has led to an epidemic of MPM in populations with occupational exposure to asbestos and those with environmental and paraoccupational exposures. Indeed, the incidence of mesothelioma is closely related to historical asbestos utilisation in the last century. MPM carries a poor prognosis with median survival generally less than one year after diagnosis. Advances in medical imaging and diagnostic techniques have improved the diagnostic process although therapeutic advances have been frustratingly slow.

## 2. Epidemiology of Mesothelioma

Malignant pleural mesothelioma was first described in 1947 [1] with a further description of “endothelioma of the pleura” in 1955 [2]. MPM was considered rare with little known about its association with asbestos exposure, until a report in 1960 of 33 cases of pleural malignancy in people with industrial exposure to blue asbestos (crocidolite) from South Africa [3].

Asbestos derives from a Greek word meaning “inextinguishable”, and it has been used for its physical and electrochemical insulating properties perhaps for as long as 5000 years [4]. Asbestos describes the group of naturally occurring fibrous silica-based minerals that are divided based on their microscopic appearance into the “needle-like” amphiboles which include crocidolite, amosite (brown asbestos), actinolite, anthophyllite and tremolite, and the “curved, feather-like” serpentine asbestos known as chrysotile (white asbestos). The widespread industrial use of asbestos in the last century mostly involved crocidolite, amosite, and chrysotile. The only other mineral associated with the occurrence of MPM is erionite, which has not been used commercially but is responsible for endemic MPM in Cappadocia, Turkey [5], where it occurs naturally in the rocks.

The true global burden of MPM is unclear, largely due to varied recording and reporting methods used across different countries. However, since the 1960s the disease has been widely recognised in asbestos-exposed populations, reflecting the previous industrial utilisation of the product in the 20th century. This is reflected by MPM mostly occurring in males with a median age around 70 years old in high-income countries (88% of deaths worldwide) [6,7,8], with almost 50,000 deaths occurring in Europe (54% of MM deaths worldwide) between 1994–2016 (see Figure 1). An estimate of the global burden of mesothelioma, utilising the World Health Organisation Mortality Database, identified 59 countries with recent quality mesothelioma data and estimated the global mesothelioma deaths to be up to 38,400 per year, based on estimates of historical asbestos use [9].

Whilst more than 60 countries have banned the import and utilisation of asbestos, the production and exportation of asbestos continues with Russia, Kazakhstan, China, Brazil and Zimbabwe producing an estimated 1,100,000 metric tonnes of asbestos in 2019 [11,12]. In turn, middle and low income countries continue to utilise asbestos with China and India estimated to consume more than half of the global supply of asbestos [13]. This therefore represents a significant on-going health hazard for many developing countries [14].

Asbestos has been mixed with tens of thousands of products including brakes and gaskets for motor vehicles, fireproofing, insulation lagging, electrical components, cement sheets, vinyl, tiles, textiles and even cigarette filters. Different “waves” of MPM incidence have been described that coincide with the exposure history of different populations: the first wave resulting from exposure in the mining, milling and transport of raw asbestos; the second wave from the use of asbestos in industry, such as manufacturing, insulation, shipping, electrical, boilermaker, railway, armed services, carpentry, engineering and cement industries; the third wave from renovations/repairs or demolition to buildings containing asbestos [8,15,16]. It remains unclear as to when the peak in incidence of mesothelioma will occur in more developed countries, with on-going trends of increasing incidence and male mortality published from a wide range of countries [17,18,19,20,21,22].

In taking an occupational exposure history, clinicians must also explore “para-occupational” exposure. This describes asbestos exposure incurred by working alongside others who were working with asbestos. This also includes exposure of household contacts of workers, where exposure of family members to asbestos dust from the overalls of tradesmen is well-recognised. In this population, there is a higher proportion of women compared to other cohorts [23]. A detailed assessment should also include any history of performing renovations to homes or buildings that may have contained asbestos-containing materials. Other individuals may have been exposed through working in non-industrial buildings that contained asbestos (for example teachers in schools), because they have lived in the vicinity of asbestos mines or factories, or even visited old industrial sites with residual environmental asbestos contamination.

The different asbestos fibres confer different risk of pleural malignancy with amphiboles (especially crocidolite) much more potent than chrysotile [24]. This is consistent with experimental data demonstrating experimentally that durable long fibres (>5μ) of small diameter are the most potent in inducing MPM [25]. The exposure-specific risk for crocidolite, amosite; and chrysotile is estimated to be 500:100:10 [25]. There is controversy as to whether chrysotile causes MPM, for instance a recent large cohort study of Dutch motor vehicle mechanics demonstrated a lower age-adjusted risk of death from mesothelioma compared to controls, perhaps due to exposure predominantly to commercial chrysotile [26]. The debate and confusion may be partly attributed to the contamination of most chrysotile deposits with amphiboles (such as tremolite) [27,28] and the fact that the different types of asbestos were mixed commercially depending on market value and availability. 

There is no known threshold below which exposure to asbestos is considered safe. There is a clear dose response relationship to exposure and risk of MPM (and lung cancer), with increasing risk over time since first exposure, the power of the exponent being between 3 and 4, at least for the first 30–40 years after first exposure [29]. Estimates of latency continue to be revised as exposed populations age; the Western Australia Mesothelioma Registry initially reported a time since first exposure to diagnosis of those diagnosed between 1960–1979 of 26 years [8], with the most recent estimate of latency in those diagnosed between 2010–2019 being 52 years [30]. This observation is, in part, not surprising as the period of highest asbestos use, and thus exposure of the population in most developed countries, is fixed in the 1960–1970′s, and as the population at risk grows older, the latency will be prolonged. These data also indicate that there does not yet seem to be a period of time beyond which the risk of developing mesothelioma falls.

While asbestos exposure is undoubtedly the most significant risk for mesothelioma, polio vaccine contamination with Simian virus 40 (a DNA monkey virus) between 1950 and 1970 has also been associated with MPM [31], although it is more likely a co-carcinogen with asbestos [32]. Radiation has also been implicated as a potential cause of MPM [33,34,35,36,37,38]. Familial risk has been described in individuals with a first or second-degree relative with MPM having an approximately two-fold increase in rate of MPM, even after accounting for the degree of asbestos exposure [39]. Germline mutations in the BAP1 (BRCA1-associated protein 1) gene have been identified in families with a high incidence of MM [40,41]. A genome-wide association study of common variants in MPM identified several candidate-gene regions [42], although further work is required to replicate and extend these findings. 

## 3. Histopathology

MPM first develops on the parietal pleura where it may appear as multiple small grape-like nodules. The nodules coalesce to form more of a continuous sheet of tumour with gradual progression to involve the visceral pleura. Macroscopically, focal necrosis or haemorrhage may be seen. Encasement of the lung and other structures in the thorax may occur as a layer of dense tissue up to several centimetres thick extending into the fissures. Metastasis is a late feature of mesothelioma, but at death tumour deposits may be widespread [43].

Mesothelioma has distinctive histological subtypes, epithelioid (the most common, up to two-thirds of cases), sarcomatoid, and biphasic (a mixture of epithelioid and sarcomatoid). These subtypes confer distinct survival properties, with epithelioid mesothelioma generally being associated with a better prognosis (see Figure 2). It is not clear what leads to or why there are different subtypes, with an analysis of more than 2000 mesothelioma cases failing to identify any association with different asbestos fibre types, latency or duration, or cumulative exposure characteristics [44].

Serum or pleural fluid biomarkers (e.g., soluble mesothelin related protein (SMRP), osteopontin, fibrilin-3) may be of use to increase the diagnostic probability of MPM [45,46,47,48,49], however, pathological material remains necessary if possible, to confirm malignancy. This may be accomplished by the cytological examination of pleural fluid, fine needle aspiration or a biopsy of the solid tumour, using an image guided biopsy approach or thoracoscopy.

Microscopically, it is necessary to differentiate MPM from other tumours (e.g., epithelioid and adenocarcinoma, sarcomatoid and other spindle cell tumours) [50], with Calretinin and WT-1 most specific for MPM when used with carcinoma markers. Specific immunohistochemistry stains which, when used in batches, have good sensitivity and specificity to define different histology more reliably [51,52]. This has reduced the use of electron microscopy (to identify particular ultrastructural changes and distinguish sarcomatoid MPM from other spindle cell tumours) to only very rare cases. Importantly, calretinin can identify cells as being of mesothelial origin and appropriate epithelial membrane staining antigens are highly suggestive of mesothelioma [51] and assist to differentiate epithelioid MM from reactive mesothelial hyperplasia and adenocarcinoma.

Advances in the understanding of chromosomal losses in MPM have significantly transformed the diagnostic process in recent years. The routine diagnostic process should now involve testing for the inactivation of tumour suppressor genes, in particular the loss of p16INK4A-p14ARF (CDKN2A), neurofibromatosis type 2 (NF2) and BRCA1-associated protein 1 (BAP-1) [53,54,55]. Homozygous deletion of CDKN2A can be demonstrated through FISH analysis of the 9p21 region and/or it can be reliably shown by immunohistochemistry using MTAP as a surrogate marker [56]. Loss of BAP-1 may be an early finding during the development of MPM and may be seen in undiagnosed pleural effusions present years prior to proven invasive MPM, which has led to recent interest in the concept of mesothelioma-in-situ [57,58,59].

Experienced pathology laboratories can reliably diagnose MPM by cytology alone from pleural aspiration with a sample fulfilling one or more of the following criteria: indisputable malignant cells on cytomorphological criteria (demonstrating mesothelial phenotype); or cytomorphological features which are not unequivocally malignant, but with ancillary techniques such as loss of BAP-1 or p16 (CDKN2A) [60,61]. However, the use of cytology alone for the diagnosis of epithelioid MPM has been controversial due to concern over the potential for miss-attribution of a biphasic tumour as epithelioid MPM, as epithelioid cells readily shed into a pleural effusion and sarcomatoid cells do not. If this were the case then presumably there would be different survival characteristics of a population with cytologically-diagnosed epithelioid MPM, as it would also contain some biphasic mesothelioma. However, a large study of 1212 epithelioid mesothelioma cases with 499 diagnosed by cytology alone failed to demonstrate a difference in survival between cytology- and histology-diagnosed epithelioid mesothelioma, suggesting that this presumption is not true, at least when using survival as a proxy for a histological subtype [62]. Further, audits of practice have demonstrated that a cytological diagnosis is available on average 29 days earlier before a subsequent confirmatory histological report, thus significantly reducing the delay in the diagnostic process for patients and avoiding the need for more invasive diagnostic procedures [63]. Taken together, a cytological diagnosis of epithelioid mesothelioma should be regarded as reliable with appropriate cytomorphological/tumour suppressor gene features present, and this approach does not appear to have any clinically meaningful difference in overall survival when compared to a diagnosis using histology. While pleural fluid aspiration alone is highly desirable and convenient alone for some patients, it should be noted that with advances in immunotherapy there is an increasing requirement for histological samples to establish which therapeutic approaches may be appropriate and, in addition, entry into many clinical trials may require a histological confirmation of mesothelioma.

## 4. Clinical Presentation and Investigations

MPM classically presents with progressive dyspnoea, weight loss and chest wall pain, with the presence of a unilateral blood-stained pleural effusion and/or pleural thickening. Fatigue is often under-recognised, and fevers may be present [64]. Extra-pulmonary restriction with ventilatory impairment may be present (with neither obstructive nor restrictive patterns) as measured by a reduction in the forced expiratory volume in 1 s (FEV1) and forced vital capacity (FVC) with normal FEV1/FVC ratio. Total lung capacity and gas transfer are reduced with an increase in the diffusion constant, as with diffuse pleural thickening.

The plain chest X-ray (CXR) of MPM will usually show the presence of a pleural effusion or pleural thickening, which should be confirmed by contrast enhanced computed tomography (CT) (see Figure 3). Magnetic Resonance Imaging (MRI) scans can be used but rarely add much over and above CT imaging. Chest wall invasion may be seen on CT or MRI scans, thereby providing further evidence of a malignant process. Fluorodeoxyglucose positron emission tomography (FDG-PET) has been shown to be helpful in diagnosis (see Figure 4), particularly in differentiation between benign and malignant pleural disease with a standardised uptake value (SUV) below 2.2, demonstrating the highest predictive value for benign disease [65]. PET can also be used to identify the most likely biopsy sites to yield a diagnosis if pleural fluid cytology is unhelpful or ultrasound guided pleural fluid aspiration is not possible [66]. PET scans can be falsely positive in the presence of infection/other inflammatory pleural conditions and so it is important to consider these as alternative causes of “positive” PET imaging. Tumour staging by the TNM system may be assessed by CT scan [67], MRI or PET, although in most instances there is limited direct clinical utility in staging, unless radical surgery is being considered (see below).

The natural history of MPM is usually one of relentless increase in tumour size and bulk with a resultant increasing burden of symptoms (particularly chest wall pain, dyspnoea, fatigue, and weight loss) [64]. Anxiety, low mood, and anhedonia is present in around half of patients at the time of diagnosis and a low reported health related quality of life is associated with an increased risk of death [64]. Whilst the physical symptoms are mostly attributable to local pathology in the chest, a series examining 318 post-mortem results of subjects with MPM demonstrated a remarkably high rate of metastases, both within and outside of the thoracic cavity [43]. Rarely, a patient may present with metastases [68].

## 5. Treatment of Mesothelioma

The effectiveness of systemic anticancer therapy for MPM has been limited to date, although there is increasing interest and data supporting the role of immunotherapy with immune checkpoint inhibition [69,70]. Single agent chemotherapy is ineffective. Until recently, the only approved systemic anticancer therapy for MPM was platinum-based chemotherapy combined with pemetrexed [71], with or without bevacizumab [72]. Recent regulatory approvals for immunotherapy are now providing further therapeutic options. Therapy should be regularly monitored with CT or FDG-PET imaging so that it can be discontinued or changed when disease progression is apparent. 

There has been increasing interest and progress in the role of immunotherapy and checkpoint inhibition for MPM. The initial role was unclear, with programmed cell death ligand 1 (PD-L1) inhibitors demonstrating only modest response rates between 10 and 29% in phase II trials with variable progression free (PFS) and overall survival (OS) figures [69]. Single agent pembrolizumab was not superior to chemotherapy in a recent multicentre randomised controlled trial [73]. A combination of durvalumab (anti-PD-L1) with cisplatin-pemetrexed as first line has shown some promise in a single centre phase II study [74], with recruitment into a larger phase III study on-going. 

The recent report from the Checkmate 743 study, an open label, multicentre randomised controlled study, describes the interim analysis of 605 patients with unresectable MPM randomly assigned to have first line nivolumab (PD-L1 blocking) plus ipilimumab (anti-cytotoxic T-lymphocyte 4 (CTLA-4) versus standard care of platinum plus pemetrexed chemotherapy [75]. The headline results of this study demonstrated a statistically significant improvement in the primary endpoint of OS, with the median OS of 14.1 months for patients who received chemotherapy, versus 18.1 months for those who received dual immunotherapy (HR 0.74, 96·6% CI 0·60–0·91). The most remarkable aspect of this study is that a priori sub analyses demonstrates the driver for the benefit in OS is almost entirely in those with non-epithelioid subtype. The median OS of patients with non-epithelioid disease was 8.8 months (95% CI 7.4–10.2) for those receiving chemotherapy versus 16.5 months (95% CI 14.9–20.5) for those receiving dual immunotherapy (HR 0.46, 95% CI 0.31–0.69) [75]. Hitherto, a non-epithelioid subtype has been associated with a worse outcome and treatment resistance [76,77] (see prognosis section below). It is already known that the sarcomatoid subtype generally has a higher PD-L1 expression [78] than epithelioid, although in Checkmate 743 a cut point of ≥1% seemed to have less influence than histology subtype on the OS [75]. There remains an urgent requirement to elucidate clinically useful and prognostic biomarkers to inform treatment strategies for MPM.

Malignant pleural effusion is very common in MPM and is associated with a worse quality of life. In recent years the objectives of managing a malignant effusion has begun to focus more on patient centred outcomes such as health related quality of life and around reduction, or improvement of, breathlessness rather than radiological appearances alone. Control of a malignant pleural effusion (and thus reduced dyspnoea) may be achieved by pleurodesis using talc insufflation or slurry (or other sclerosing agent) through a pleural drain, or with an indwelling pleural drainage catheter (IPC) [79], or both with IPC and talc slurry [80]. The presence of trapped lung indicates that attempts at pleurodesis will be futile, in which case an IPC alone is indicated. The AMPLE study was a multicentre randomised controlled trial that demonstrated that patients with malignant pleural effusion spend less time in hospital after IPC placement than those with a chest drain and attempt at pleurodesis, but there was no difference in objective dyspnoea scores, quality of life or survival [81]. The subsequent AMPLE-2 randomised controlled study demonstrated that patients with an IPC having daily drainage reported no difference in dyspnoea scores but were more likely to achieve a pleurodesis [82].

Radiotherapy has limited use in the treatment of MPM and is usually reserved for control of pain from catheter tract metastases through the chest wall or direct invasion of tissues such as vertebral bodies. The risk of developing intervention site metastases is likely to be related to the size of the incision for instrumentation into the pleura [83], although two large well conducted randomised controlled studies demonstrated a similar conclusion, that prophylactic radiotherapy to biopsy and intervention sites to reduce the occurrence of catheter tract metastases is not routinely indicated [84,85].

Surgical excision of MPM remains highly contentious with a lack of randomised controlled trials supporting its use. Two distinct approaches are utilised with extra-pleural pneumonectomy (EPP) consisting of an en-bloc resection of the ipsilateral lung, pleura, pericardium and diaphragm, or pleurectomy/decortication being a lung sparing approach [86]. Surgical case series have reports of survival of up to 30% at five years in highly selected populations (reviewed in [86]). The only two randomised controlled trials published to date have failed to demonstrate any survival benefit [87,88], either by EPP or by video-assisted partial pleurectomy, with adjuvant chemotherapy and/or radiotherapy [88,89]. The complications of surgery and adjunct therapy are significant [86] and, at present, there is no agreed established role for radical surgery in managing MPM [90] with widespread acceptance that such advanced operations should only be conducted in high volume specialist surgical centres, ideally as part of clinical trials [91]. The role of palliative surgery in managing trapped lung or malignant pleural effusion is less controversial, although the randomised MESOVATS study comparing a video-assisted thoracoscopic surgery approach for partial pleurectomy against chest drain talc slurry failed to demonstrate any survival benefit and surgery was associated with more complications and longer hospital stay [92].

For many patients, symptom palliation alone is arguably the most important aspect of management of MPM although clinical trials are yet to establish the best timing and nature of this support [93]. The RESPECT-Meso study was an international multicentre randomised controlled study examining the utility of regular early specialist palliative care for patients recently diagnosed with MPM. The primary endpoint was not met with no significant between group differences in health-related quality of life twelve weeks after diagnosis [93]. It is clear that specialist palliative care has a highly significant role in many patient journeys, and it is likely that the RESPECT-Meso study recruited patients too early in their disease trajectory for the intervention to have a clinically meaningful impact on the outcome measures. Multidisciplinary advanced care planning and palliative care of the terminally ill patient and their family and/or dependents is recognised as increasingly important [64,94]. A randomised study of nurse-led conversation on end-of-life care compared to standard care demonstrated the initiation of such a conversation is both acceptable and more likely to result in an advanced health directive for those with severe respiratory disease [94]. Medicolegal compensation relating to past asbestos occupational exposure often greatly concerns many patients and their families and clinicians should advise patients of the potential for legal redress, dependent on local jurisdictional legal statutes.

## 6. Prognosis

The prognosis of MPM is poor with a median survival between 9 and 12 months, although rare cases of long survival are recognised [95]. Population-based studies consistently confirm that non-epithelioid histology, increasing age and male gender are independent risk factors for poor outcome for pleural MM [96,97]. Peritoneal mesothelioma has similar prognostic factors to pleural [98] and generally has a worse prognosis than pleural mesothelioma. Biomarkers such as SMRP, osteopontin and fibrilin-3 provide some prognostic information [49,99,100,101,102] with definite limitations. Genetic and immunohistochemical studies continue to provide advances in the understanding of tumour biology but with no clear clinically useful validated prognostic factors for MPM identified to date. Nuclear mitotic and atypia grading systems may provide useful prognostic knowledge [103] but require further evaluation. Modern metabolic imaging techniques such as PET/CT can provide some prognostic information using baseline total glycolytic volumes [77].

Inflammation affects the development and progression of many cancers and the release of proinflammatory cytokines in MPM may be associated with systemic inflammatory symptoms, such as fever, sweating, and weight loss. The neutrophil-to-lymphocyte ratio (NLR) is a marker of systemic inflammation that can be calculated from a differential white cell count, with a high ratio suggesting greater systemic inflammation. Worse outcomes have been demonstrated in two separate MPM populations undergoing systemic anticancer therapy (ratio ≥5 associated with worse hazard ratio for survival) and extra-pleural pneumonectomy (ratio ≥3 associated with a worse prognosis) [104,105].

Many clinically based prognostic scoring systems have mainly been based on highly selected clinical trial populations, selected for their fitness for attempts at surgical treatment and/or chemotherapy [106,107,108]. Consequently, these are not generalisable and have not been widely utilised. The most consistently reported variables associated with a poor prognosis have been older age [95,96,109,110,111,112,113], male sex [95,110,112] and sarcomatoid histological characteristics [95,96,109,111,112,113]. A recent prognostic model in MPM patients who had undergone surgery sought to combine tumour characteristics and patient phenotype, reporting that tumour volume, molecular expression subtype, NLR, Eastern Cooperative Oncology Group performance status (ECOG PS) and serum albumin were all associated with survival [106].

Probably the most clinically useful prediction model is based on routinely available clinical parameters at the time of diagnosis [76,114]. This classification and regression tree model uses combinations of clinical variables present at the time of diagnosis (weight loss, haemoglobin, performance score, albumin, histology) to categorise patients with MPM into different risk groups, with distinct survival differences. For example, the combination of weight loss with ECOG PS of 2 or more, or weight loss, ECOG PS 0–1 with sarcomatoid histology, conferred the worst median (interquartile range: IQR) survival of 7.4 (3.3 to 11.1) months, whereas the combination of no weight loss, normal haemoglobin and normal serum albumin was associated with the best survival with a median (IQR) 34.0 (22.9–47.0) months [76]. The model has been validated in different populations of MPM patients and may help stratify risk groups to inform different treatment options and discussions with patients and their families [115,116].

## 7. Conclusions

Historical trends in asbestos utilisation in the last century continue to drive the incidence of mesothelioma, although asbestos mining and manufacture continues in many countries today. In developed countries, residual asbestos containing material in the built environment represents a continuing risk of exposure and this is likely to lead to a long tail of the mesothelioma epidemic in decades to come. Advances in the diagnostic process has led to earlier diagnosis and perhaps an increasing lead-time bias for survival. Treatment options remain limited, and survival is poor with modest responses to systemic anticancer therapy to date, albeit with some promising recent advances in immunotherapy, particularly in individuals with sarcomatoid histology who hitherto have had a worse outcome. Palliative approaches to identify and manage physical and emotional symptoms remain a priority for many patients with mesothelioma.

## Figures and Tables

**Figure 1 cancers-13-04194-f001:**
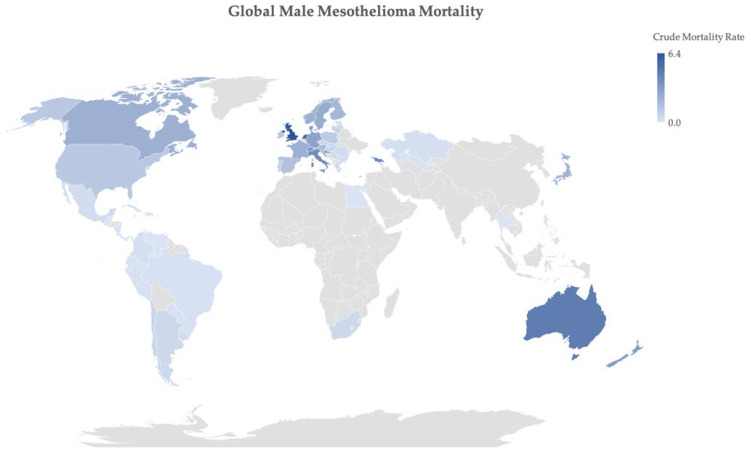
Global male mesothelioma mortality up to 2017, expressed as crude rate per 100,000; source: World Health Organization mortality database [10].

**Figure 2 cancers-13-04194-f002:**
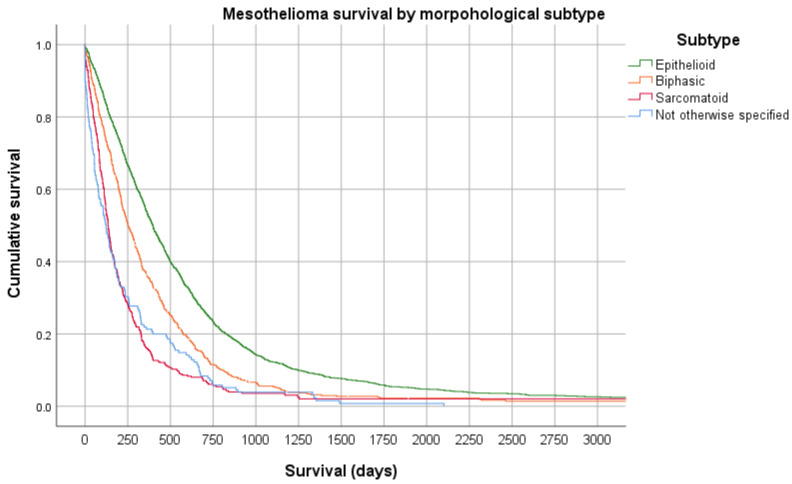
Kaplan Meier survival curve of 2796 cases of mesothelioma from the Western Australia Mesothelioma Registry stratified by morphological subtype demonstrating a significant difference in overall survival between subtypes (Log Rank *p* < 0.0001).

**Figure 3 cancers-13-04194-f003:**
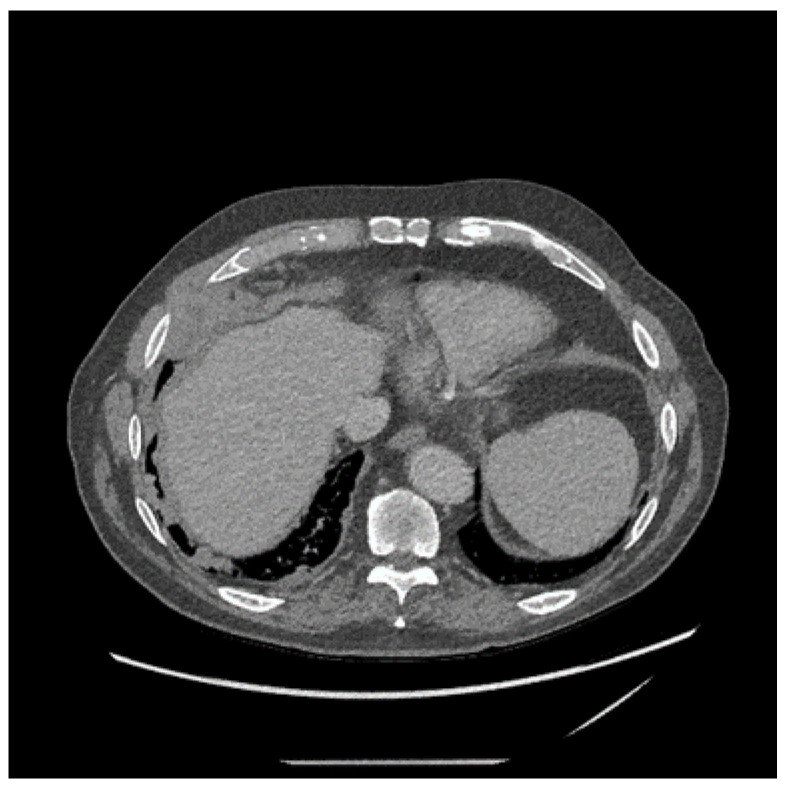
Non-contrast enhanced computed tomography of the chest in a 79-year-old man involved in the mining of crocidolite demonstrating nodular thickening of the pleura in the right base. Image guided fine need aspiration confirmed the presence of epithelioid mesothelioma.

**Figure 4 cancers-13-04194-f004:**
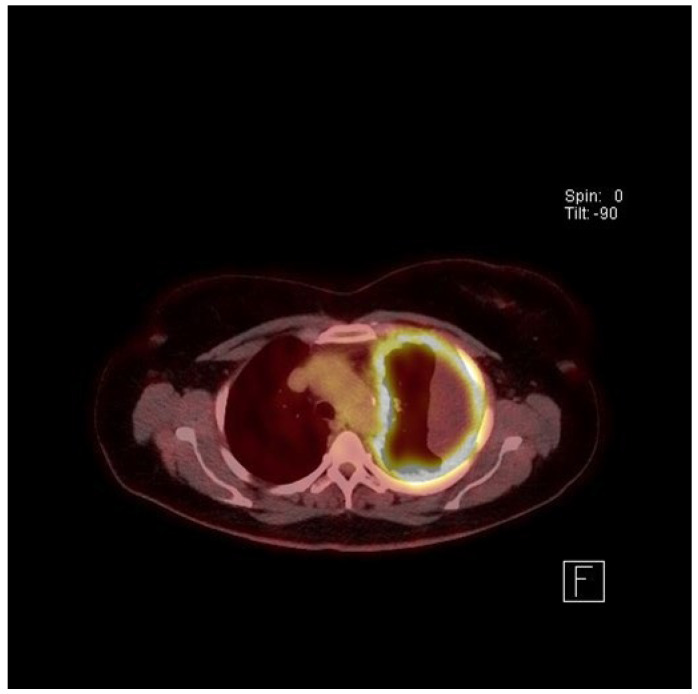
FDG−PET of a 68-year-old female whose husband worked as a carpenter, demonstrating circumferential pleural thickening with increased FDG activity and loculated malignant pleural effusion.

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
