# Peer review of "Epidemiology and Clinical Aspects of Malignant Pleural Mesothelioma"

_cancers, 2021, doi:10.3390/cancers13164194_

Round 1
Reviewer 1 Report
The manuscript "Epidemiology and clinical aspects of malignant pleural mesothelioma" is well-written and interesting-to-read, and provides a current overview of the epidemiological and clinical aspects of mesothelioma. My comments are as follows:
Major comment:
"Chapter 2. Epidemiology of mesothelioma" warrants, in my opinion, expansion to include epidemiology on the current global asbestos production and use, so that the reader familiarises with the fact that asbestos is an ongoing health hazard for many countries, especially overpopulated developing ones, and therefore mesothelioma incidence is globally increasing. The author makes a mention of this only in his conclusion, but it is important to elaborate in the text.
Minor comments:
-Page 3, lines 79-88: in this paragraph, there also needs to be mentioned environmental exposure within specific professions who work in non-industrial buildings that contain asbestos, like teachers in schools.
-Page 5, line 145: please replace "highly desirable" with "necessary, if possible". Diagnostic sampling may differ from country to country and depend on patient clinical presentation, however it is recognised that tissue is the gold standard for mesothelioma diagnosis.
-Page 8, line 266: please delete extra space in front of "1%"
Author Response
Production and utilisation - new para page 3, lines 67-73
All other minor suggestions amended
Many thanks
Reviewer 2 Report
The author provides an excellent and clear overview regarding the history of MPM and its correlation with asbestos usage as well as the current clinical management of that disease. This work may also be a good introduction for any beginner in the field of mesothelioma research.
I recommend this review for publication
Author Response
Thank you fot your kind comments
Reviewer 3 Report
Dear Authors:
The authors have carried out the study “ Epidemiology and clinical aspects of malignant pleural mesothelioma”. The aim of this study is to describe the knowledge on the epidemiology of mesothelioma and provide an overview of best clinical practice including diagnostic approaches and management of this daunting disease.
Some considerations need to be taken into account:
This manuscript is a fine, interesting and well designed review of the disease.
From an academic point of view the manuscript is original and gives good knowledge and current update of malignant pleural mesotelioma.
References should be described as recommended by the style guide of the journal. Journals should be cited as an Abbreviated Journal Name. Please review references num 7,8,10,34,36,45,66,71,74,76,78,81,87 and 102.
It would have been desirable to use more recent bibliographic references (especially from 2018 to the present) because their percentage is low.
Only 19 out of 113 references have been used from 2018 to the present. Some very recent and important useful references are attached for this review among others as an example:
- Intraoperative argon-plasma coagulation treatment for patients with malignant pleural mesothelioma. Iyoda A et al. Mol Clin Oncol. 2021;15(3):188.
- Heterogeneity of treatment effects in malignant pleural mesothelioma. Di Maio M et al.Lancet. 2021 24;398(10297):301-302.
- Malignant Mesothelioma. Pouliquen DL et al. J. Cancers (Basel). 20219;13(14):3447.
- Immunotherapy in malignant pleural mesothelioma: a review of literature data. Menis J et al.Transl Lung Cancer Res. 2021;10(6):2988-3000.
- Immunotherapeutic Approaches in Malignant Pleural Mesothelioma. Terenziani R et al. Cancers (Basel). 2021 4;13(11):2793.
- Immune checkpoint inhibitors a new player in the therapeutic game of mesothelioma: New reality with new challenges. Parikh K et al. J.Cancer Treat Rev. 2021 16;99:102250.
- Update on Diagnosing and Reporting Malignant Pleural Mesothelioma. Savic I et al. J.Acta Med Acad. 2021 Apr;50(1):197-208.
- Oncological Frontiers in the Treatment of Malignant Pleural Mesothelioma. Vita E et al. E.J Clin Med. 2021 25;10(11):2290.
- Epidemiology, diagnosis and treatment of malignant pleural mesothelioma, a narrative review of literature. Schumann SO et al. J Thorac Dis. 2021;13(4):2510-2523.
Kind regards
Author Response
Refernce style - I have used Endnote with the latest MDPI citation requirements. I will need advice as to appropriate journal abreviations if that is what is required.
Recent references. I have added 2 more recent refs. The other refs suggested below represent recent reviews and editorials - not original work or research. I have worked hard to ensure that the original reference and data are cited otherwise.